# Optimal Sizing of Movable Energy Resources for Enhanced Resilience in Distribution Systems: A Techno-Economic Analysis

**Mukesh Gautam** [1,*] and **Mohammed Ben-Idris** [2]

1 Power & Energy Systems Department, Idaho National Laboratory, Idaho Falls, ID 83415, USA
2 Department of Electrical and Computer Engineering, Michigan State University, East Lansing, MI 48824, USA; benidris@msu.edu
* Correspondence: mukesh.gautam@nevada.unr.edu

**Abstract:** This article introduces a techno-economic analysis aimed at identifying the optimal total size of movable energy resources (MERs) to enhance the resilience of electric power supply. The core focus of this approach is to determine the total size of MERs required within the distribution network to expedite restoration after extreme events. Leveraging distribution line fragility curves, the proposed methodology generates numerous line outage scenarios, with scenario reduction techniques employed to minimize computational burden. For each reduced multiple line outage scenario, a systematic reconfiguration of the distribution network, represented as a graph, is executed using tie-switches within the system. To evaluate each locational combination of MERs for a specific number of these resources, the expected load curtailment (ELC) is calculated by summing the load curtailment within microgrids formed due to multiple line outages. This process is repeated for all possible locational combinations of MERs to determine minimal ELC for each MER total size. For every MER total size, the minimal ELCs are determined. Finally, a techno-economic analysis is performed using power outage cost and investment cost of MERs to pinpoint an optimal total size of MERs for the distribution system. To demonstrate the effectiveness of the proposed approach, case studies are conducted on the 33-node and the modified IEEE 123-node distribution test systems.

**Keywords:** distribution system resilience; expected load curtailment; graph theory; movable energy resources; techno-economic analysis

## 1. Introduction

The frequency of extreme events, both natural (e.g., hurricanes, wildfires, earthquakes, and windstorms) and man-made (e.g., cyber and physical attacks), has increased significantly over the recent decades [1]. In the United States, there has been a noticeable rise in the frequency of outages caused by weather-related extreme events between 1992 and 2012, according to research conducted by the U.S. Energy Information Administration [2]. Figure 1 shows the frequency of weather-related billion-dollar extreme events that have occurred in the United States from 1980 to 2022 based on the data collected by National Oceanic and Atmospheric Administration [3]. In the year 2022, the United States experienced 18 severe weather-related disasters, each resulting in economic damages exceeding one billion dollars. These events have caused destruction to major power system components and subsequently grid-wide prolonged power disruptions. Failures of power distribution system components (contribute to about 90% outages in the United States [4]) are major causes of outages to a significant number of customers [5]. Severe weather events and their subsequent power outages have posed a significant challenge to electric utilities in fulfilling their mission of providing reliable and resilient electricity services to customers. To mitigate the impact of these disruptions on end-users, strategies for power distribution system restoration (PDSR) are essential. The primary objective of PDSR is to minimize load disruptions and outage durations by optimizing available resources. Smart grid

technologies, including microgrid deployment, network reconfiguration, efficient repair crew dispatch, distributed generation resources, energy storage systems, movable energy resources (MERs), and combinations thereof, have emerged as highly effective solutions for PDSR in such scenarios.

MERs encompass several fundamental features that render them indispensable in enhancing distribution system resilience. These versatile assets possess the crucial ability to be rapidly relocated from one location to another, allowing for rapid deployment to fault locations, which is especially valuable in mitigating the impact of non-coincidental outage patterns across widespread areas [6]. MERs are not only mobile but also scalable, offering the flexibility to be configured to variable sizes, thereby catering to diverse load requirements [7]. They play a pivotal role in ensuring uninterrupted power supply to critical loads, especially in islanded distribution systems when conventional resources are unavailable or compromised due to equipment failures or damages [8].

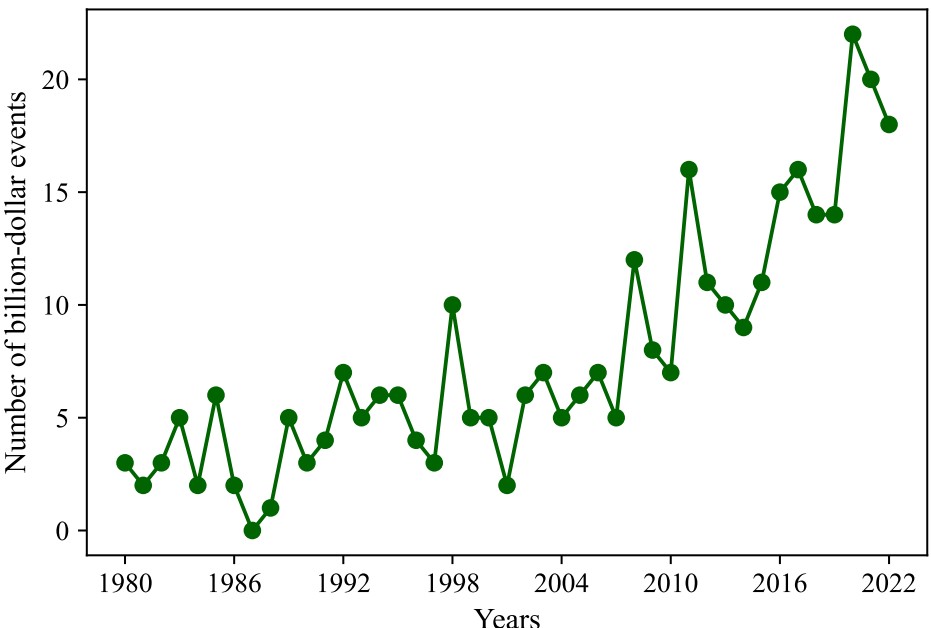

**Figure 1.** Frequency of weather-related extreme events in the United States.

The utilization of MERs for PDSR has witnessed a notable surge in interest and implementation. To improve the resilience of distribution systems, a robust two-stage optimization technique is introduced in [9]. This technique focuses on the scheduling and routing of MERs. Similarly, in the pursuit of enhancing seismic resilience, a two-stage PDSR approach based on Mixed-Integer Linear Programming (MILP) is presented in [10], tailored specifically for systems with MERs. Furthermore, a PDSR strategy based on MILP is put forth in [11] for active distribution systems. This strategy coordinates the scheduling and routing of mobile energy storage systems to enhance resilience. Another facet of PDSR is showcased in [12], where an optimization technique is presented to restore unbalanced distribution systems effectively. This technique exhibits the versatility to coordinate diverse sources of distributed generating resources, encompassing both renewable and non-renewable types, in conjunction with battery energy storage systems. Additionally, ref. [12]includes the optimization of mobile generator dispatch, optimizing their contribution to resilience enhancement. In the pursuit of enhancing distribution system resilience with the integration of mobile energy storage systems, a two-stage optimization strategy is introduced in [13]. This strategy encompasses dynamic microgrid formation, offering a multifaceted approach to resilience enhancement. Moreover, in [14], a concept of separable mobile energy storage systems is introduced. This approach aims to broaden the horizons of MER applications by enhancing their adaptability. The scheduling constraints for MERs are meticulously derived, taking into account their intricate interactions with the

distribution system. In [15], an optimal decision support system is presented to assist microgrid operators in managing various emergencies, including extreme weather and technical issues, by maximizing the autonomy of the microgrid and prioritizing renewable energy sources. In [16], an emergency power supply recovery strategy for distribution networks with microgrid support is investigated to address uncertainties caused by renewable energy sources, demonstrating its effectiveness in improving energy safety and stabilty in such networks. A robust approach is presented in [17] to coordinate repair and dispatch resources, including renewable energy sources, to maximize the restored load in a distribution system after extreme weather events while ensuring resilience against uncertainties, ultimately enhancing the efficiency of service restoration. Despite the substantial progress in the field of PDSR, as evidenced by the emergence of various optimization strategies for routing, scheduling, and coordination of MERs, a noticeable research gap remains. The majority of the above literature primarily concentrates on the coordination and dispatch of MERs alongside other PDSR techniques for enhanced distribution system resilience. However, a critical aspect often left unaddressed is the optimal sizing of MERs, specifically tailored to enhance distribution system resilience. This omission underscores the need for an approach that not only considers the deployment and utilization of MERs but also assesses their size in the context of distribution system resilience enhancement. This research bridges this existing gap by introducing a comprehensive techno-economic framework for determining the optimal total size of MERs, thereby contributing a crucial dimension to the evolving field of PDSR.

In our previous research [18], the focus was to determine the optimal number and total size of MERs based on technical criteria. In this article, a techno-economic criterion for determining the optimal total size of MERs is introduced. We continue to use high wind speed as an example of extreme weather events, which aligns with our earlier study [18]. Within this framework, a wide range of multiple line outage instances is generated utilizing forecasted wind speeds, leveraging distribution line fragility curves. Scenario reduction is performed on this diverse set of scenarios applying the fuzzy k-means algorithm. These reduced scenarios form the basis for assessing expected load curtailments (ELCs) contingent on the strategic placement of MERs at various network nodes. The reduction in scenarios plays a pivotal role in this process, minimizing computational complexity while preserving the core attributes of the original dataset. Importantly, each reduced scenario requires the reconfiguration of the distribution network, which is modeled as a graph, enabling systematic evaluation. The ELC of each locational combination of MERs is then calculated for a given number of MERs. For any MER total size, the minimal ELCs are determined. Case studies are conducted using a 33-node system and a modified IEEE 123-node system to demonstrate the practical utility and promising potential of the proposed approach in enhancing distribution system resilience. The main contributions of this article are as follows:

- Introduction of a comprehensive techno-economic framework to determine the optimal total size of MERs to enhance distribution system resilience.
- Employing the fuzzy k-means algorithm for efficient scenario reduction, reducing computational complexity while preserving key attributes of original scenarios.
- Conduction of practical case studies using the 33-node and 123-node systems to showcase the real-world applicability of the proposed approach.
- Comparison of technical and techno-economic criteria, consistently showing that the latter results in cost-effective solutions for distribution system resilience.

The remainder of the article is structured as follows: Section 2 elaborates on the fundamental mathematical modeling ideas that form the foundation of this research on optimizing MER sizes. In Section 3, we delve into the details of our proposed approach and present the solution algorithm. To demonstrate the application of our approach, case studies are conducted on both 33-node and modified IEEE 123-node systems in Section 4. The discussion of the case study results are presented in Section 5. Finally, in Section 6,

our findings and contributions are summarized, offering concluding remarks and insights we gained.

## 2. Mathematical Modeling

In this section, we introduce some key mathematical ideas that form the foundation of our research. First, we explore graph theory, which helps us create models of distribution networks. Then, we explain graph-related concepts like "spanning trees" and "spanning forests". These concepts are crucial for us to figure out how to fit MERs perfectly into resilient and reliable distribution networks. These mathematical tools are the keys to finding the optimal total size of MERs.

### 2.1. Graph Theory

Within the domain of mathematics, graph theory is the discipline devoted to exploring, constructing, and scrutinizing graphs. In this context, a graph serves as a structured arrangement of objects, where certain pairs of these objects are conceptually linked. These objects are symbolized through mathematical entities referred to as vertices (also known as nodes or points), and every connection between pairs of vertices is designated as an edge (alternatively described as a link or line) [19]. Graphs are conventionally depicted visually, often as an assembly of dots or circles signifying the vertices, accompanied by lines or curves symbolizing the edges. It is important to note that these edges can take on two forms: they may either be directed or undirected. From a mathematical standpoint, a graph can be formally represented as a pair denoted as $G = (N, E)$, where $N$ constitutes a set encompassing entities referred to as nodes, and $E$ signifies a set of nodes that are interconnected, termed edges.

The size of a graph is contingent upon the quantity of nodes it contains. Within a graph, a path denotes a route that can be traversed by following edges and passing through nodes. Every component of a path, including its edges and nodes, maintains a connection with one another. Conversely, a cycle, sometimes referred to as a circuit, represents a specific type of path that commences and concludes at the very same node. The length of a path or cycle is precisely determined by the count of its edges. When there exists a path connecting each pair of nodes within a graph, it is classified as a connected graph [20]. Within the realm of connected graphs, there is a special type known as a tree, characterized by its lack of cycles. In the context of a tree graph featuring $|N|$ nodes and $|E|$ edges, this particular relationship is expressed by equation below:

$$|N| = |E| - 1. \tag{1}$$

### 2.2. Graph Representation of Distribution Grid

In distribution system networks, there exist two types of switches: sectionalizing switches (typically in a closed state) and tie-switches (typically in an open state). When all switches within a distribution network are in the closed position, it results in the formation of a meshed network. This meshed network can be effectively depicted as an undirected graph, denoted as $\mathscr{G} = (\mathscr{N}, \mathscr{E})$, wherein $\mathscr{N}$ represents a collection of nodes (also known as vertices), and $\mathscr{E}$ signifies a collection of edges (also referred to as branches).

#### 2.2.1. Spanning Tree

A "spanning tree" is a fundamental concept closely associated with connected graphs. Understanding the concept of a spanning tree is essential for comprehending more complex graph structures. A *spanning tree* of the undirected graph $\mathscr{G} = (\mathscr{N}, \mathscr{E})$ is essentially a subgraph that includes all the vertices of the original graph $\mathscr{G}$ and forms a tree structure, which means it is acyclic (no cycles) and connected (there is a path between any two vertices). Spanning trees are used in graph theory and network analysis to understand the structure and connectivity of a graph while minimizing the number of edges needed to connect all the vertices [21]. In a connected graph, there exist multiple spanning trees,

and each of these trees shares an equal count of edges and vertices. Within the undirected graph $\mathscr{G}$, each edge is assigned a distinct numerical value, often referred to as weights, and these weights can vary depending on the specific problem at hand. The sum of these edge weights is minimized when the minimum spanning tree is constructed. In Figure 2a, we can observe a spanning tree within an imaginary 12-node system. This particular spanning tree, depicted in the figure, encompasses all the nodes within the system, which amounts to a total of 12 nodes, and it includes 11 closed branches or edges.

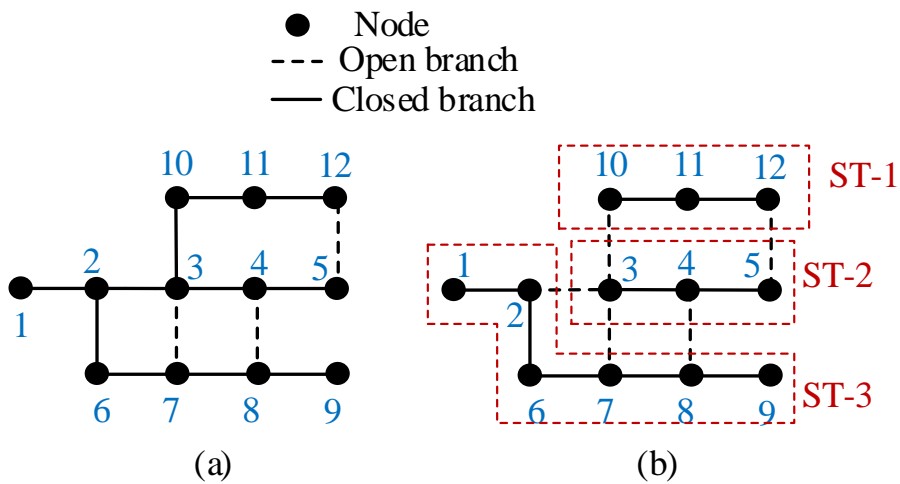

**Figure 2.** (**a**) A spanning tree and (**b**) a spanning forest of an imaginary 12-node system.

2.2.2. Spanning Forest

In the realm of graph theory, the concept of a "spanning forest" is intimately tied to undirected graphs. To understand the concept of a spanning forest, it is essential to first understand the concept of a "forest" in graph theory. A *forest* is a collection of *trees*, where each tree comprises a set of nodes connected by edges, and no two trees within the forest are interconnected. Essentially, a forest is a graph that contains multiple isolated trees, and there are no cycles within each tree. Now, when discussing a *spanning forest*, we are referring to a specific scenario within an undirected graph $\mathscr{G}$. In this context, a spanning forest encompasses all the nodes of the graph but is composed of a collection of distinct, non-connected spanning trees [21]. In scenarios where all the spanning trees are interconnected, every node within the undirected graph $\mathscr{G}$ is incorporated into one of these spanning trees, as emphasized in [22]. Conversely, when a disconnected graph comprises multiple linked segments, it assembles a spanning forest, encompassing a spanning tree for each of these individual components, as highlighted in [23]. In Figure 2b, we can observe a spanning forest that formed due to removal of two other edges (2–3 and 3–10) in the previously depicted spanning tree in Figure 2a. This particular spanning forest, depicted in Figure 2b, is composed of three distinct spanning trees (ST-1, ST-2, and ST-3).

Kruskal's spanning forest search algorithm (KSFSA) [24] is employed in this work to seek out the spanning forest. KSFSA commences by forming an initial forest denoted as $F$, where each node within the graph is considered an individual tree. Operating in a greedy fashion, KSFSA then proceeds to link the next least-weighted edge that does not create loops or cycles within the existing forest $F$ during each iteration. Ultimately, the resultant forest $F$ after the final iteration stands as the optimal spanning forest. Figure 3 illustrates the process through the flowchart of KSFSA.

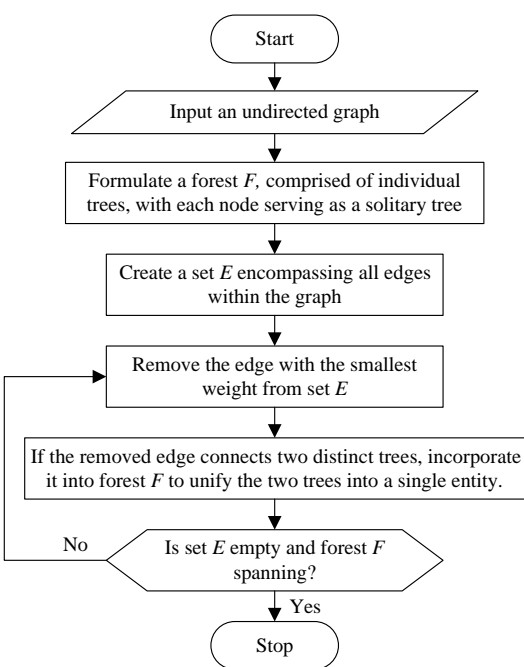

**Figure 3.** Flowchart depicting Kruskal's algorithm to search for spanning forest [25].

## 3. Proposed Approach

In this section, we explore our proposed approach, which involves several critical stages. This comprehensive methodology includes extreme event modeling, scenario generation and reduction, calculating expected load curtailment, and a techno-economic analysis to find the optimal total size of MERs in a distribution system.

### 3.1. Generating Line Outage Scenarios in Extreme Weather with Fragility Curves

In this article, weather-related fragility curves are employed as a tool to simulate extreme events and create numerous scenarios involving line outages. These fragility curves serve as a means to depict how different components within the system perform and their susceptibilities when faced with unpredictable extreme weather conditions. To determine the failure probabilities of different components, weather forecast data are utilized to map them onto the fragility curve, as detailed in [26]. For the purpose of this study, the concept is illustrated with the example of multiple line outages triggered by high wind speeds, which is a prime illustration of an extreme weather event. The likelihood of line failure induced by high wind speeds is mathematically expressed as follows [27]:

$$P_l(w) = \begin{cases} \overline{P_l}, & \text{if } w < w_{\text{crl}} \\ P_{l\_hw}(w), & \text{if } w_{\text{crl}} \leq w < w_{\text{cpse}} \\ 1, & \text{if } w \geq w_{\text{cpse}} \end{cases}, \tag{2}$$

where $P_l$ represents the probability of line failure, and this probability is influenced by the wind speed $w$; $\overline{P_l}$ signifies the failure probability under standard weather conditions; $P_{l\_hw}$ represents the line failure probability when faced with high winds; $w_{\text{crl}}$ is the critical wind speed, meaning the threshold at which distribution lines begin to experience failures; and $w_{\text{cpse}}$ is the speed when distribution lines entirely collapse.

### 3.2. Fuzzy k-Means Scenario Reduction

While mimicking a substantial number of multiple line outage scenarios can undoubtedly enhance the accuracy of an approach, handling such an extensive dataset can be computationally challenging and time-consuming. This is where the fuzzy k-means algorithm comes into play. Fuzzy k-means is a form of soft clustering that permits scenarios to

belong to multiple reduced clusters, thereby introducing a degree of fuzziness and overlap between these clusters [28]. By incorporating this technique, the computational complexity of the proposed approach is effectively streamlined while retaining its essential features intact. This allows striking a balance between precision and computational efficiency.

Let us consider an initial collection of scenarios, denoted as $\mathcal{X} = \{x_1, \ldots, x_r\}$, and $\mathcal{M} = \{\mu_1, \ldots, \mu_K\}$ as the set of reduced scenarios, often referred to as cluster centroids. If we define the extent to which any data point $x_i$ from $\mathcal{X}$ belongs to the $j$th scenario cluster with a weight $u_{ji}$, then the cluster centroid of the $j$th reduced scenario can be determined by calculating the weighted average of all the original scenarios. The mathematical expression to calculate the cluster centroid can be represented as follows:

$$\mu_j = \frac{\sum\limits_{i=1}^{r} u_{ji}^m \times x_i}{\sum\limits_{i=1}^{r} u_{ji}^m}. \tag{3}$$

In (3), $m \in [1, \infty)$ is the hyperparameter, referred to as fuzzifier, that plays a crucial role in shaping the level of fuzziness within the clusters [29]. The value of $m$ controls the level of of influence that each data point has on the assignment of multiple clusters. When $m$ is set to 1, the clustering algorithm behaves more like traditional hard clustering (i.e., k-means clustering), where data points are distinctly assigned to a single cluster with no overlap. As $m$ increases, the influence of each data point on multiple clusters becomes more pronounced, resulting in softer, more overlapping cluster assignments.

To arrive at the final values of cluster centroids, we engage in an iterative minimization process of the objective Function (4), as detailed in [30].

$$\min \sum_{i=1}^{r} \sum_{j=1}^{K} u_{ji}^m ||x_i - \mu_j||^2, \tag{4}$$

where

$$u_{ji} = \frac{1}{\sum\limits_{k=1}^{K} \left( \frac{||x_i - \mu_j||}{||x_i - \mu_k||} \right)^{\frac{2}{m-1}}}. \tag{5}$$

To assess the quality of the scenario reduction process, a comprehensive evaluation is conducted, comparing the effectiveness and quality of the fuzzy k-means algorithm with that of the k-mean and k-median algorithms. Our assessment relies on a set of important metrics, including the Silhouette (SL) index, the Calinski–Harabasz (CH) index, and the Davies–Bouldin (DB) index.

The SL index, one of our primary evaluation metrics, plays a crucial role assessing the quality of scenario reduction. It serves as a yardstick to measure how well an original scenario aligns with its own cluster when contrasted with other clusters. Operating on a scale from −1 to +1, the SL index provides valuable insights into the degree of alignment. A higher SL index signifies a strong alignment between the scenario and its designated cluster, whereas a lower value suggests a weaker alignment with other clusters. Mathematically, the SL index is expressed as shown in (6), which enables us to quantitatively evaluate the alignment of scenarios within their respective clusters and their distinctiveness from scenarios in other clusters [31].

$$S_L = \frac{1}{r} \sum_{i=1}^{r} \left( \frac{b_i - a_i}{\max\{a_i, b_i\}} \right). \tag{6}$$

Each element in (6) is defined as follows: $a_i$ represents the mean separation between the $i$th scenario and the remaining scenarios belonging to the same cluster, indicating the degree of cohesiveness within the cluster. Conversely, $b_i$ represents the shortest separation between

the $i$th scenario and scenarios found in different clusters, indicating the degree of separation or dissimilarity between clusters.

The CH index assesses how widely clusters are distributed from each other. It quantifies the ratio of dispersion between clusters to dispersion within clusters [32]. This index is also known as the variance ratio index, and higher CH values signify superior clustering. Mathematically, the CH index is represented as follows [32]:

$$CH = \frac{B_K \times (r - K)}{W_K \times (K - 1)}, \tag{7}$$

where $B_K$ represents inter-cluster covariance and $W_K$ represents intra-cluster covariance.

The DB index employs the built-in qualities and attributes of data to assess the clustering performance. It achieves this by comparing the mean similarity of each cluster with respect to its nearest neighbor. Here, similarity signifies the ratio of distances within a cluster to distances between clusters [33]. Consequently, clusters that are more uniformly distributed receive higher scores. The DB index ranges from a minimum value of 0, where lower values signify better clustering. The mathematical expression for the DB index is given by (8) shown below [33]:

$$DB = \frac{1}{K} \sum_{j=1}^{K} \max_{i \neq j} \frac{S_j + S_i}{M_{ji}}, \tag{8}$$

where $S_j$ is a metric for quantifying the distance within the $j$th cluster, and $M_{ji}$ is a metric for measuring the separation between clusters $j$ and $i$.

### 3.3. Computing Expected Load Curtailment (ELC)

The ELC refers to the expected value of the total amount of the curtailed critical loads. Since the amounts of curtailed critical loads are different for different outage scenarios, the ELC is calculated to obtain an expected value of load curtailment out of all reduced scenarios. It reflects the weighted average of load curtailments across multiple scenarios, considering their likelihood of occurrence.

The ELC for the $i$th locational combination is calculated by summing the product of the probability of every reduced scenario and the critical load curtailment for that scenario, as shown in (9).

$$ELC_i = \sum_{j=1}^{M} Pr(j) \times LC_i(j), \tag{9}$$

where $M$ denotes the total count of reduced scenarios obtained after applying the fuzzy k-means algorithm; $Pr(j)$ denotes the probability of the $j$th reduced scenario; and $LC_i(j)$ represents the amount of curtailed critical load of the $j$th reduced scenario for locational combination $i$.

The critical load curtailment is calculated as the sum of the load curtailments at each node in the system, weighted by the critical laod factor. Mathematically, the critical load curtailment is represented by (10) shown below:

$$LC_i(j) = \sum_{x=1}^{N} \omega_x \Delta P_{xi}(j), \tag{10}$$

where $\Delta P_{xi}(j)$ denotes the load curtailment at node $x$ of the $j$th reduced scenario for locational combination $i$; $\omega_x$ denotes the critical load level at node $x$; and $N$ signifies the total number of system nodes.

When calculating the critical load curtailment, it is essential to ensure compliance with node power balance constraints and the radiality constraint, as outlined below.

*(a) Nodal power balance constraints:* The nodal power balance constraint ensures that the sum of power injected from sources, including MERs, and the power flowing through

lines at each node equals the load demand of the node. The balance of power at each node within the system is represented as follows:

$$\sum_{k \in \Omega_g(k)} P_{g,k} + \sum_{b \in \Omega_B(k)} P_{b,k} = P_{D,k}. \tag{11}$$

In (11), $\Omega_g(k)$ represents the set of generating sources, including MER, linked to node $k$; $\Omega_B(k)$ represents the set of branches connected to node $k$; $P_{g,k}$ signifies the power supplied by generating source $k$; $P_{D,k}$ indicates the power demand at node $k$; and $P_{b,k}$ represents the branch power flow from node $b$ to node $k$.

*(b) Radiality constraint:* A fundamental requirement for a distribution system is radiality, meaning that each potential configuration must adhere to this constraint. In essence, the radiality constraint demands that every spanning tree within the network maintains a tree-like or radial structure. The radiality constraint in a power distribution system can be described mathematically using the node-branch incidence matrix and the concept of Kirchoff's Current Law (KCL). The node-branch incidence matrix is used to describe how branches are connected to nodes in any network.

Let us suppose that each spanning tree is modeled as a sub-graph $\mathscr{G}_s = (\mathscr{N}_s, \mathscr{E}_s)$, where $\mathscr{N}_s$ constitutes the set of nodes, and $\mathscr{E}_s$ encompasses the set of branches within this sub-graph. If we denote $n_s$ as the count of nodes and $e_s$ as the count of branches within a specific spanning tree, the node–branch incidence matrix $A \in \mathbb{R}^{n_s \times e_s}$ is constructed with its elements, $a_{ij}$, determined according to (12). Specifically, $a_{ij}$ is assigned the following values: +1 if branch $j$ originates at node $i$, $-1$ if branch $j$ terminates at node $i$, and 0 otherwise.

$$a_{ij} = \begin{cases} +1 & \text{if branch } j \text{ starts at node } i \\ -1 & \text{if branch } j \text{ ends at node } i \\ 0 & \text{otherwise} \end{cases}. \tag{12}$$

For a distribution system to be radial, it should satisfy the following conditions:

- *Single Root Node*: There should be one designated root node (usually the substation denoted by $0_s$) from which all other nodes receive power. This can be represented as follows:

$$\sum_{j \in \mathscr{E}_s} a_{ij} = 0, \forall i \in \mathscr{N}_s \setminus \{0_s\}. \tag{13}$$

In this equation, the sum of incoming branches to all nodes (except the root node $0_s$) should be zero indicating all power comes from the root node.

- *No Loops or Cycles*: There should be no closed loops or cycles in the network. This contraint can be expressed as follows:

$$\sum_{i \in \mathscr{N}_s} a_{ij} \leq 1, \forall j \in \mathscr{E}_s. \tag{14}$$

This equation ensures that each branch has either one or zero nodes connected to it. If a branch has more than one node connected to it, it implies the existence of a loop.

### 3.4. Determination of Optimum Total Size of MERs

Based on the optimum number of MERs, a techno-economic analysis is performed to determine the optimum total size of MERs for a distribution system. While performing the techno-economic analysis, the total of two types of costs, i.e., power outage cost and investment cost is computed to select the optimum total size of MERs. The power outage cost $C_{outage}$ is calculated for each total MER size based on (15).

$$C_{outage} = ELC_{min} \times t_{outage} \times VoLL, \tag{15}$$

where $ELC_{min}$ is the minimum expected load curtailment for a particular total MER size $P_{MER-tot}$; $t_{outage}$ is the duration of power outage; and $VoLL$ is the value of lost load (VoLL).

For the calculation of the investment cost of MERs, the levelized cost of electricity (LCOE) of MER is considered in addition to backup time requirement and the total size of MERs. The investment cost of MERs $C_{investment}$ is calculated for total MER size based on (16).

$$C_{investment} = P_{MER-tot} \times LCOE_{MER} \times t_{backup}, \tag{16}$$

where $LCOE_{MER}$ is LCOE of the backup generator considered for MERs; and $t_{backup}$ is the backup duration requirement.

The Levelized Cost of Electricity (LCOE), also referred to as levelized cost of energy, is the net present worth of electricity generated by a power plant or an electricity generated averaged over its lifetime [34]. It is commonly used for investment decision making and comparing different sources of electricity. It is the ratio of the discounted total cost of constructing and operating a power plant over the course of its lifetime to the discounted value of the actual energy that can be produced over the lifetime of the power plant. The LCOE can alternatively be thought of as the lowest price that the energy should be sold for the power plant to break even during its lifetime [34]. Mathematically, the LCOE can be calculated as follows [35]:

$$LCOE = \frac{\sum\limits_{t=0}^{T} \frac{C_t}{(1+r)^t}}{\sum\limits_{t=0}^{T} \frac{E_t}{(1+r)^t}}, \tag{17}$$

where $C_t$ denotes net expenditure that includes capital cost, operation and maintenance cost (O and M), and fuel costs (if applicable) in year $t$; $E_t$ denotes the actual energy generated in year $t$; $T$ denotes total assumed lifetime; and $r$ denotes discount rate.

The flowchart in Figure 4 shows the overall steps to determine the optimal total size of MERs using the proposed solution algorithm.

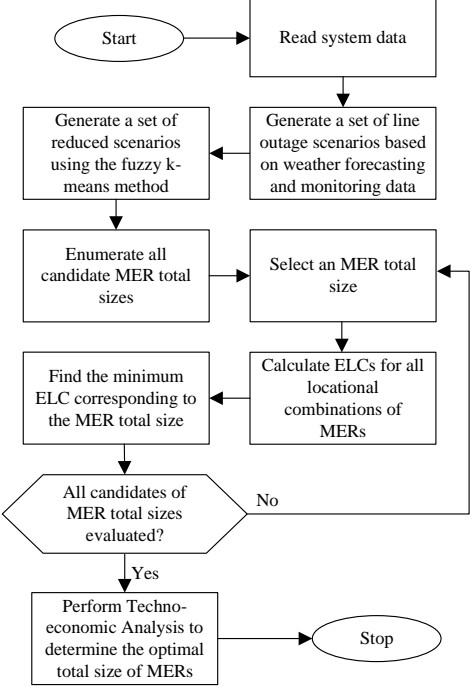

**Figure 4.** Flowchart of the proposed solution algorithm to determine the optimal total size of MERs.

## 4. Numerical Analysis and Results

In this section, the applicability and effectiveness of the proposed approach is showcased by conducting numerical analyses on two diverse distribution systems: the 33-node and the IEEE 123-node systems. These numerical studies help us illustrate how our proposed methodology works in practice and its impact on optimal MER sizing.

### 4.1. Systems Descriptions

This study relies on numerical simulations conducted using the 33-node system and a modified IEEE 123-node system.

The 33-node distribution test system [36] is a widely recognized benchmark system used for research and analysis in the field of power distribution. It is characterized by a radial configuration, meaning it follows a tree-like structure. The key attributes of the 33-node system are as follows:

- *Topology*: This system comprises 33 nodes, 32 branches, and five tie-lines, resulting in a total of 37 branches. The system's branches, including the tie-lines, are uniquely numbered from 1 to 37 for the purpose of analysis as depicted in Figure 5.
- *Aggregate Load*: The aggregate load within the 33-node system is approximately 3.71 MW. The total load is distributed across different system nodes.
- *Critical Loads*: To evaluate the system performance during extreme events, specific critical loads are considered within the 33-node system. The amounts and locations of critical loads within the 33-node system are shown in Table A1 of Appendix A. The total amount of critical loads in this system is 1265 kW and the distribution of critical loads across various nodes are indicated in Figure 5.

The modified IEEE 123-node system serves as another testbed for our research and analysis. It shares similarities with the 33-node system in terms of radial configuration and critical load considerations but offers a larger and more complex network for investigation. The key characteristics of the modified IEEE 123-node system are as follows:

- *Topology*: This system consists of 123 nodes and 126 branches, which form a radial distribution network as depicted in Figure 6. Similar to the 33-node system, it adheres to a tree-like structure.
- *Branches with Tie-Switches*: Within the network, two branches (specifically branches 94–54 and 151–300) are equipped with tie-switches, allowing for controlled reconfiguration during system disturbances.
- *Aggregate Load*: The aggregate load within the IEEE 123-node system is approximately 3.47 MW. The total load is distributed across different system nodes.
- *Load Balance*: All branches and loads within the modified IEEE 123-node system are assumed to be balanced, simplifying the analysis while maintaining a realistic representation of a distribution system.
- *Critical Loads*: As in the case of the 33-node system, specific critical loads within this system are identified. The critical load details for this system are shown in Table A2 of Appendix A. The total amount of critical loads in this system is 815 kW and the distribution of critical loads across various nodes are indicated in Figure 6.

### 4.2. Implementation and Results in Case of the 33-Node System

To implement the proposed approach for determining the optimal total size of MERs in the case of the 33-node system, multiple line outage scenarios are generated using a high wind speed condition as a representative extreme weather event. In this study, the critical wind speed and the collapse speed of 30 m/s and 55 m/s, respectively, are assumed as parameters associated with the fragility model (2) [27]. During normal weather conditions, a failure probability of 0.01 is considered. The failure probability starts to increase after reaching 30 m/s and follows a linear pattern up to 55 m/s. The resulting wind fragility curve for distribution lines is depicted in Figure 7. To create a robust dataset for analysis, 10,000 random outage scenarios are generated.

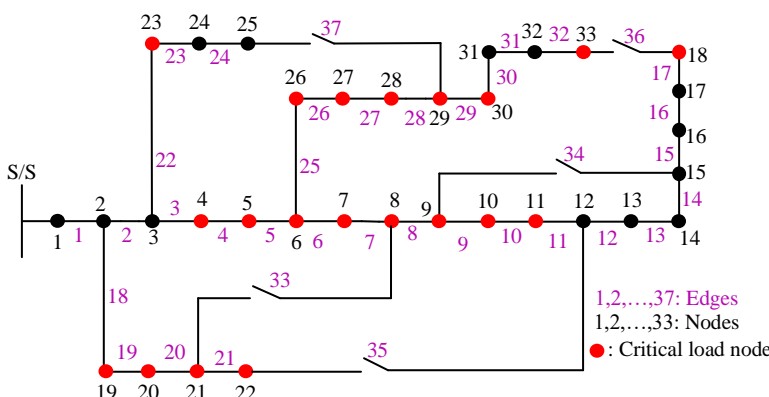

**Figure 5.** 33-node distribution test system.

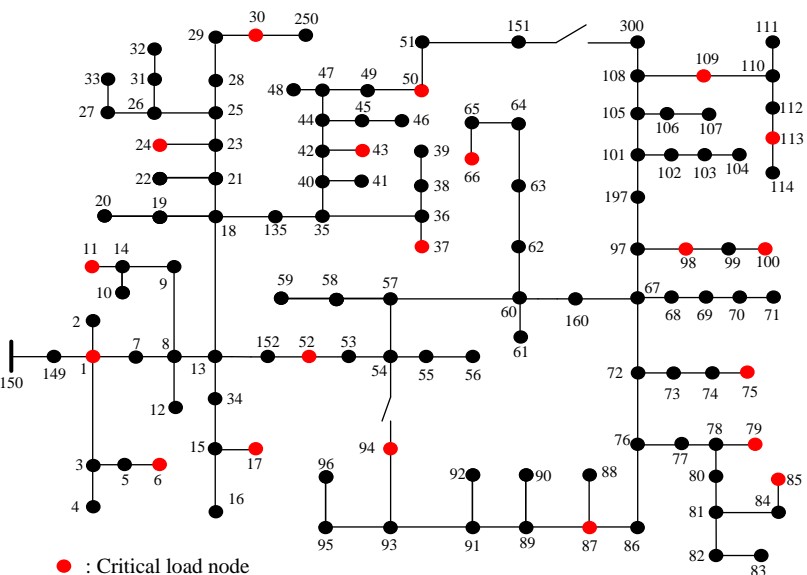

**Figure 6.** Modified IEEE 123-node system.

Subsequently, the fuzzy k-means algorithm is employed to effectively reduce the initially generated scenarios into a more manageable set of 200 reduced outage scenarios, all occurring at wind speeds of 38 m/s. This application of the fuzzy k-means algorithm results in 200 reduced line outage scenarios, each associated with its respective probability of occurrence. To assess the quality of our scenario reduction approach, it is compared with alternative clustering algorithms, specifically k-mean [37] and k-median [38] algorithms. To conduct this comparison, three evaluation indices, namely the Silhouette (SL) index, the Calinski–Harabasz (CH) index, and the Davies–Bouldin (DB) index, are employed. These indices help gauge the effectiveness of the fuzzy k-means algorithm in relation to the other clustering techniques.

Table 1 provides a summary of the index values for all three clustering algorithms. The results of this comparison reveal that the fuzzy k-means algorithm outperforms the other algorithms in terms of all three indices, indicating its superior performance in scenario reduction.

**Table 1.** Comparative Assessment of Scenario Reduction for the 33-node System.

| Index/Algorithm | k-Means | k-Medians | Fuzzy k-Means |
|---|---|---|---|
| SL index | 0.0257 | 0.0119 | 0.0292 |
| CH index | 19.507 | 16.947 | 20.211 |
| DB index | 2.869 | 3.116 | 2.833 |

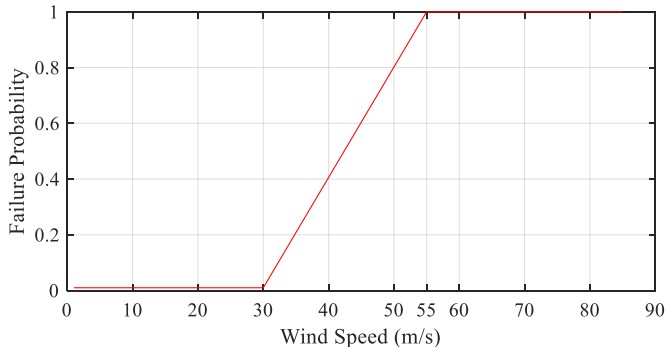

**Figure 7.** Distribution Line Wind Fragility Curve.

The total sizes of MERs ranging from 500 kW to 1900 kW are taken at a granularity level of 100 kW. The methodology outlined in our previous work [18] is adopted to determine the optimal number of MERs, which is seven for the 33-node system. The individual sizes of MERs are determined by dividing the total MER size by the optimal number of MERs. For each candidate total MER size, there are multiple locational combinations. Seven MERs could be distributed on various nodes using a number of ways. For each locational combination of MERs, the expected load curtailment (ELC) is determined by considering critical load curtailments corresponding to each of the 200 reduced line outage scenarios and their probabilities. Based on these ELC values, the minimum ELC for each MER total size is calculated.

*Results of Techno-Economic Criteria*: For techno-economic analysis, the sum of power outage cost (15) and investment cost (16) is used to determine the optimum total size of MERs. The value of lost load, i.e., *VoLL* in (15) is taken as USD 10/kWh. The outage time and backup time in (15) and (16) are both set equal to 72 h based on backup duration suggested by Federal Emergency Management Agency (FEMA) for a long-term power outage [39]. The LCOE of MER is calculated using the LCOE calculator [40] developed by National Renewable Energy Laboratory (NREL). During the calculation of LCOE, the MER is assumed to be a mobile diesel generator and the calculated value of $LCOE_{MER}$ used for the analysis is USD 0.6/kWh.

Figure 8 shows the plot of total cost as a function of total MER size based on techno-economic analysis. The investment cost is the increasing function of the total MER size, whereas the power outage cost is the decreasing function of the total MER size. When both the costs are added together, the total cost initially decreases, reaches a minimum at the total MER size of 1300 kW, and starts increasing. Therefore, the optimal total MER size is 1300 kW for the case of the 33-node system based on techno-economic analysis.

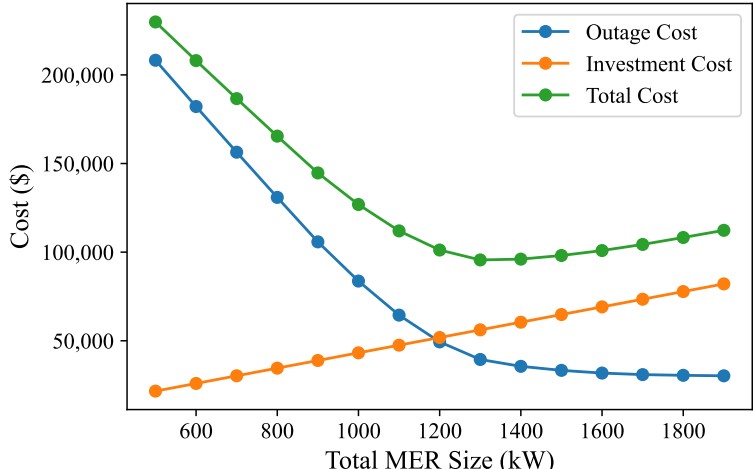

**Figure 8.** Plot of techno-economic analysis for the case of the 33-node system.

### 4.3. Implementation and Results in Case of the IEEE 123-Node System

In the context of the IEEE 123-node system, a case study approach similar to the previous one is replicated. Here, as well, 10,000 random outage scenarios are generated, and the fuzzy k-means algorithm is employed to reduce them down to 200 scenarios, each accompanied by its respective probability distribution. For a comparative assessment, three distinct indices are utilized. Table 2 presents a comparison between the fuzzy k-means algorithm and alternative scenario reduction techniques based on these indices. The analysis reveals that the fuzzy k-means algorithm outperforms the alternative algorithms in terms of scenario reduction effectiveness. The methodology outlined in our previous work [18] is adopted to determine the optimal number of MERs, which is eight for the modified IEEE 123-node system.

*Results of Techno-Economic Criteria*: Figure 9 shows the plot of total cost as a function of total MER size based on techno-economic analysis. The investment cost is the increasing function of the total MER size, whereas the power outage cost is the decreasing function of the total MER size. When both the costs are added together, the total cost initially decreases, reaches a minimum at the total MER size of 700 kW, and starts increasing. Therefore, the optimal total MER size is 700 kW for the case of IEEE 123-node system based on techno-economic analysis.

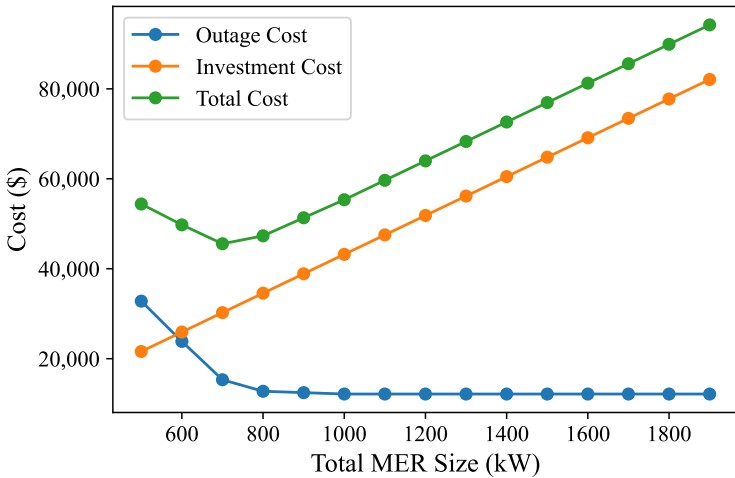

**Figure 9.** Plot of techno-economic analysis for the case of the IEEE 123-node system.

**Table 2.** Comparative Assessment of Scenario Reduction for the 123-node System.

| Index/Algorithm | k-Means | k-Medians | Fuzzy k-Means |
|:---:|:---:|:---:|:---:|
| SL index | −0.006 | −0.001 | 0.010 |
| CH index | 5.851 | 5.262 | 6.597 |
| DB index | 4.523 | 4.919 | 4.415 |

*4.4. Comparison*

To compare the results of the techno-economic analysis for both the 33-node and IEEE 123-node systems, they were evaluated against a technical criterion based on the sensitivity of the minimum ELC with respect to the total MER size, as proposed in [18].

For the 33-node system, a total MER size of 1200 kW was suggested by the technical criterion, while a total MER size of 1300 kW was indicated by the techno-economic criterion of this study. This difference in MER sizing resulted in total costs of USD 101,200 and USD 95,607, respectively, when considering both outage costs and investment costs. In contrast, for the IEEE 123-node system, both criteria converged on a total MER size of 700 kW, resulting in identical total costs of USD 45,554, considering both technical and techno-economic criteria.

Overall, the techno-economic criterion was consistently associated with either lower or equal total costs compared to the technical criterion, as illustrated in the bar chart in Figure 10. This comparison highlights the advantages of the proposed techno-economic criterion in achieving cost-effective solutions for enhancing power supply resilience in both test distribution systems.

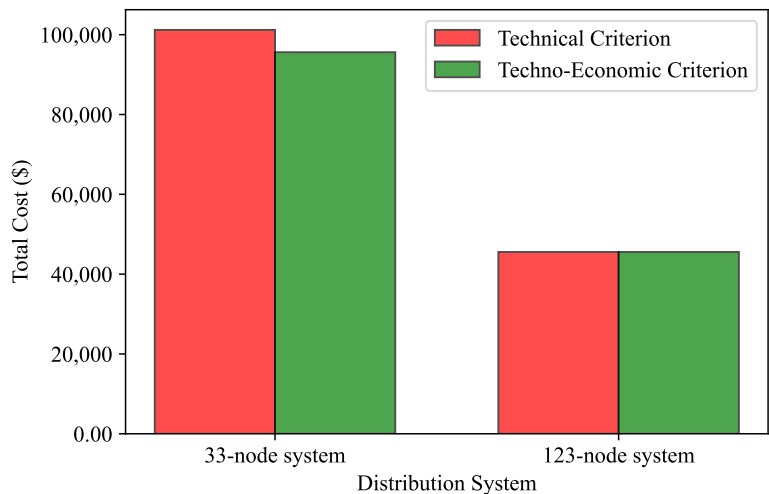

**Figure 10.** Total Cost Comparison between Technical and Techno-Economic Criteria in both Test Distribution Systems.

*4.5. Sensitivity Analysis*

To assess the sensitivity of both the total MER size and the total cost in relation to outage duration and the Value of Lost Load (VoLL), a series of sensitivity analyses were conducted.

4.5.1. Outage Duration Sensitivity Analysis

The first sensitivity analysis focuses on the impact of changes in outage duration on the total MER size and total cost for both the 33-node and IEEE 123-node systems. Outage duration was systematically varied from 24 h (equivalent to a 1-day outage) to 168 h (equivalent to a 1-week outage).

Figures [11] and [12] present the outcomes of this sensitivity analysis for the 33-node and IEEE 123-node systems, respectively. As evident from the figures, an increase in outage duration leads to a corresponding rise in total cost, which aligns with expectations. However, intriguingly, the figures also demonstrate that the total MER size remains constant throughout the varied outage durations. This observed constancy in the total MER size may seem counterintuitive, as it might be expected that longer outages would necessitate larger MER sizes for an effective resilience strategy.

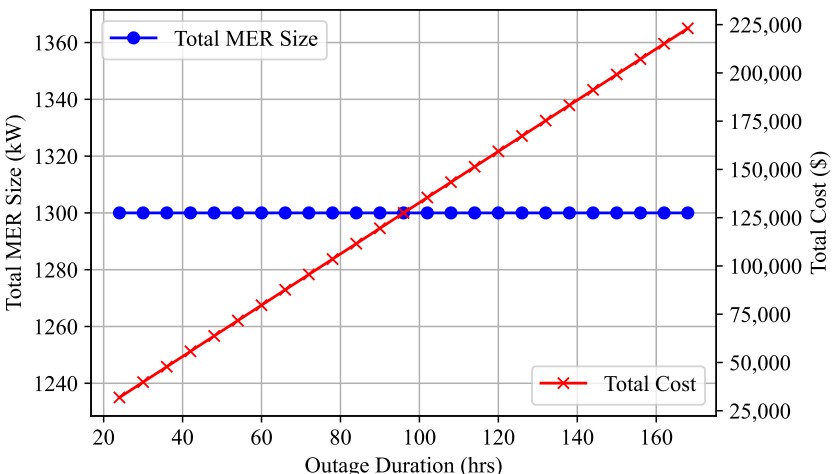

**Figure 11.** Sensitivity Analysis of Total MER Size and Total Cost with Outage Duration for the case of the 33-node system.

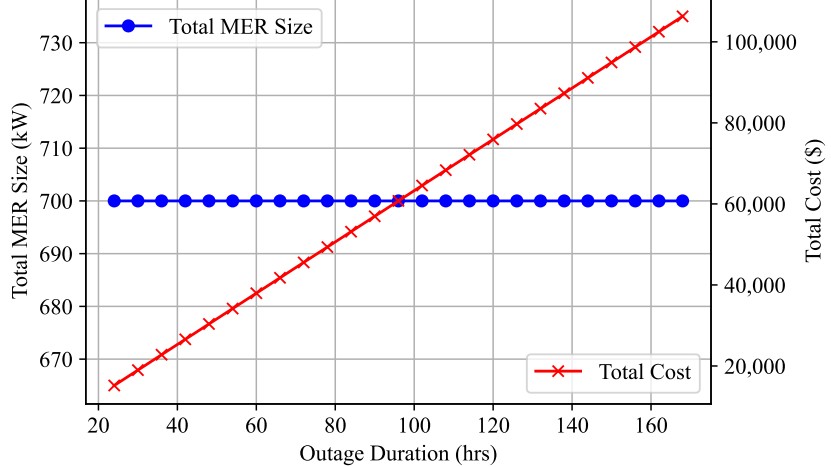

**Figure 12.** Sensitivity Analysis of Total MER Size and Total Cost with Outage Cost for the case of the IEEE 123-node system.

This constancy in the total MER size can be attributed to the underlying assumption that VoLL remains unchanged regardless of the outage duration. However, empirical research indicates that VoLL can indeed vary with outage duration, as indicated in references [41,42]. To account for this variability, an additional sensitivity analysis is required, wherein VoLL is considered as a variable of sensitivity.

### 4.5.2. VoLL Sensitivity Analysis

In the second sensitivity analysis, the VoLL parameter is varied from USD 1/kWh to USD 20/kWh, and the optimal total MER size and total cost are determined for each VoLL value. Figures [13] and [14] illustrate the results of this sensitivity analysis for the 33-node and IEEE 123-node systems, respectively. These figures distinctly illustrate that the total cost

consistently increases with higher VoLL values. However, the behavior of the total MER size exhibits a somewhat discontinuous pattern, with periods of constancy within specific VoLL ranges.

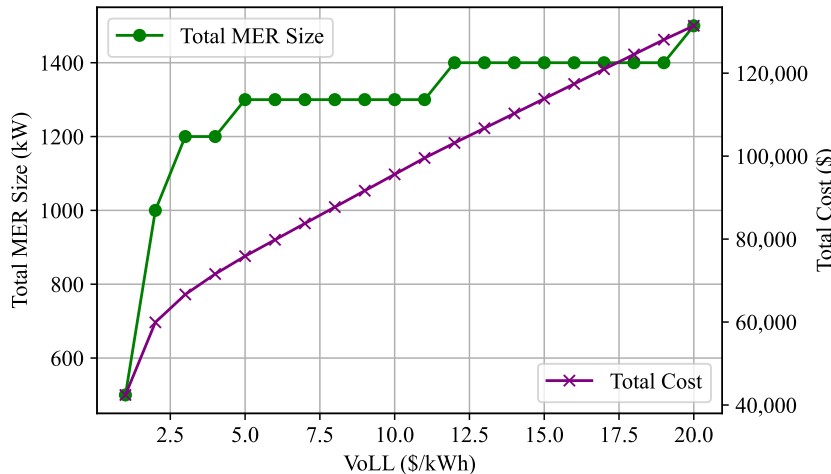

**Figure 13.** Sensitivity Analysis of Total MER Size and Total Cost with VoLL for the case of the 33-node system.

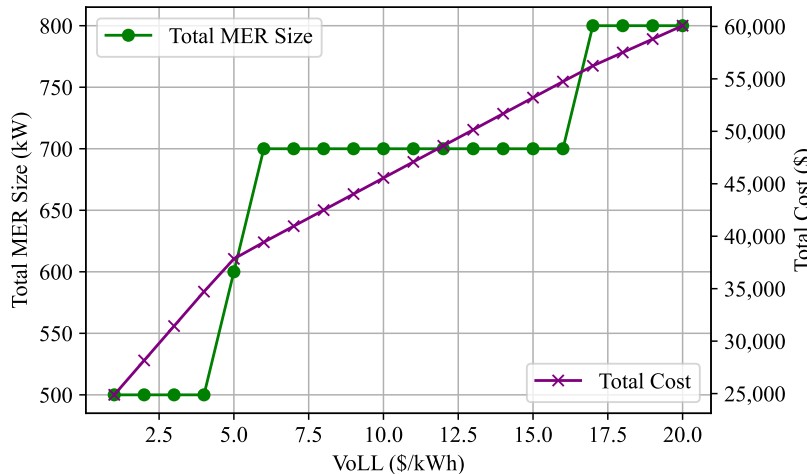

**Figure 14.** Sensitivity Analysis of Total MER Size and Total Cost with VoLL for the case of the IEEE 123-node system.

Typically, VoLL values considered for outages lasting from 24 to 72 h fall within the range of USD 5/kWh to USD 15/kWh. Notably, the VoLL value of USD 10/kWh utilized in this study to determine the optimal total MER size aligns with this typical range.

## 5. Discussion

In this section, a comprehensive discussion is undertaken regarding the findings, with a focus on the results of several key aspects of the study. These include the outcomes of the fuzzy k-means scenario reduction, the insights gained from the techno-economic analysis, a comparison of the findings with technical criteria, and a detailed exploration of the results from the sensitivity analyses. Through these facets, the aim is to illuminate the implications of the research for enhancing power supply resilience in distribution systems.

### 5.1. Fuzzy k-Means Scenario Reduction

The application of the fuzzy k-means algorithm for scenario reduction yielded promising results in both the 33-node and IEEE 123-node systems. As shown in Tables 1 and 2,

the fuzzy k-means algorithm consistently outperformed alternative clustering techniques, including k-means and k-medians, as evidenced by higher Silhouette (SL) indices, Calinski–Harabasz (CH) indices, and lower Davies–Bouldin (DB) indices. These findings demonstrate the effectiveness of fuzzy k-means in reducing the complexity of the initial scenario dataset while preserving crucial information regarding outage probabilities and their respective scenarios.

The superiority of fuzzy k-means can be attributed to its ability to capture the inherent uncertainty and overlap in outage scenarios, a feature that distinguishes it from traditional clustering methods. By assigning membership degrees to each data point, fuzzy k-means accommodates the possibility of scenarios having multiple cluster memberships, which is often the case in practical power system problems. This enhanced representation of scenario uncertainty contributes to more accurate and robust resilience planning.

### 5.2. Techno-Economic Analysis

The techno-economic analysis provides valuable insights into the determination of the optimal total size of MERs. By considering both outage costs and investment costs, this approach offers a comprehensive perspective on the trade-offs involved in enhancing power supply resilience. The results, as illustrated in Figures 8 and 9, demonstrate that the optimal total MER size depends on both outage costs and investment costs. These costs, in turn, are contingent upon specific characteristics inherent to distribution systems, such as network topology and the quantity and distribution of critical loads.

Furthermore, the techno-economic analysis reveals that the optimal total size of MERs within a distribution system is also influenced by the quantity of critical loads present in the system. Specifically, the total amounts of critical loads in the 33-node and 123-node systems are 1265 kW and 815 kW, respectively. Remarkably, this aligns with the optimal MER sizes obtained for these two systems, which were 1300 kW and 700 kW. Additionally, the optimal total size is impacted by the system's topology and the distribution of critical loads; these subtleties become evident only through meticulous analysis.

### 5.3. Comparison with Technical Criteria

The comparison between technical and techno-economic criteria, as presented in Figure 10, reveals a critical aspect of resilience planning. In cases where the technical criterion alone is considered, there is a potential risk of suboptimal MER sizing, leading to higher total costs. The techno-economic criterion consistently results in either lower or equal total costs, emphasizing its effectiveness in achieving cost-effective solutions.

In the case of the 33-node system, the techno-economic criterion identified an optimal total MER size of 1300 kW, resulting in a total cost of USD 95,607. This finding contrasts with the technical criterion, which suggested a total MER size of 1200 kW and a total cost of USD 101,200. This discrepancy underscores the importance of considering economic factors when determining optimal MER sizes, as it accounts for the cost effectiveness of larger MER capacities in reducing outage costs.

In contrast, the IEEE 123-node system yielded consistent results between the technical and techno-economic criteria, with both suggesting a total MER size of 700 kW and a total cost of USD 45,554. This congruence highlights the robustness of the techno-economic criterion in providing cost-effective resilience solutions for diverse distribution systems.

This observation underscores the necessity of integrating economic considerations into resilience planning. While technical criteria are essential for ensuring the desired level of resilience, they must be complemented by techno-economic assessments to strike an optimal balance between MER size, investment costs, and outage costs. The bar chart serves as a compelling visual representation of the tangible benefits of adopting a techno-economic perspective in distribution system resilience planning.

*5.4. Sensitivity Analysis*

The sensitivity analyses conducted for outage duration and VoLL provide deeper insights into the dynamic nature of MER sizing and cost implications. The observed constancy in the total MER size with varying outage durations, as shown in Figures 11 and 12, highlights a critical assumption that VoLL remains constant across different outage durations. However, empirical evidence suggests that VoLL can vary with the duration of power outages. This finding emphasizes the importance of revisiting the assumption of constant VoLL and conducting further research to capture the dynamics of VoLL with varying outage durations.

The VoLL sensitivity analysis, depicted in Figures 13 and 14, demonstrates the direct relationship between VoLL and total cost. Higher VoLL values result in increased total costs, which aligns with economic principles. However, the behavior of the total MER size in response to VoLL variations is more intricate. The discontinuous pattern suggests that there may be specific VoLL thresholds where further increases do not significantly impact the optimal MER size.

The sensitivity analyses highlight the necessity of more advanced approach for resilience planning in relation to VoLL and outage duration considerations. Distribution system planners can develop more accurate and cost-effective strategies for improving power supply resilience by taking into account changes in VoLL and recognizing that optimal MER sizing may not necessarily increase linearly with outage duration. It is necessary to conduct more research to better understand the relationships between these variables and to improve the proposed approach.

## 6. Conclusions and Future Work

This article presented a comprehensive approach to enhance the resilience of distribution systems through the optimal deployment of Movable Energy Resources (MERs). We introduced a techno-economic analysis that considers both power outage costs and the investment costs associated with MERs. Recognizing that this is a relatively long-term planning problem, a diverse set of line outage scenarios was generated to adequately account for uncertainties associated with extreme events. The fuzzy k-means algorithm was later employed for scenario reduction to maintain computational tractability. Due to the outage of distribution lines, the distribution network was divided into a number of microgrids and isolates. The microgrids were energized by MERs, whereas the isolates were devoid of power supply. The amount of curtailed critical loads was determined for each outage scenario.

Through extensive analysis, the expected load curtailment (ELC) was determined for various locational combinations of MERs and different MER total sizes. Our techno-economic evaluations led to the identification of optimal MER sizes. Specifically, we found that, for the 33-node and IEEE 123-node systems, optimal total sizes of 1300 kW and 700 kW, respectively, are the most cost-effective total sizes. These results demonstrate the potential of the proposed approach to enhance distribution system resilience while considering the economic feasibility of MER deployment. The total cost associated with the optimal MER size of 1300 kW in the 33-node system is USD 95,607, and for the optimal MER size of 700 kW in the IEEE 123-node system, it is USD 45,554.

Our research underscored the critical role of MERs in fortifying distribution systems against a growing frequency of extreme events, whether natural disasters or man-made disruptions. By incorporating a techno-economic perspective, we bridged the gap between technical resilience goals and cost-effective solutions. This approach allowed us the making of informed decisions regarding MER deployment, ensuring investments align with both system resilience objectives and economic feasibility.

As a future work, this research can be extended to include stability assessment. This assessment would consider the integration of MERs and explore the potential role of advanced power electronics in maintaining system stability while enhancing resilience. Additionally, further research into advanced control strategies and coordination mechanisms

for MERs in distribution systems could be pursued to enhance their effectiveness during extreme events.

**Author Contributions:** Conceptualization, M.G.; methodology, M.G.; software, M.G.; validation, M.G. and M.B.-I.; formal analysis, M.G.; investigation, M.G.; resources, M.G. and M.B.-I.; data curation, M.G.; writing—original draft preparation, M.G.; writing—review and editing, M.G. and M.B.-I.; visualization, M.G.; supervision, M.B.-I.; project administration, M.B.-I.; funding acquisition, M.B.-I. All authors have read and agreed to the published version of the manuscript.

**Funding:** This research was supported by the U.S. National Science Foundation (NSF) under Grant NSF 1847578.

**Data Availability Statement:** Not applicable.

**Conflicts of Interest:** The authors declare no conflict of interest. The funders had no role in the design of the study; in the collection, analyses, or interpretation of data; in the writing of the manuscript; or in the decision to publish the results.

## Nomenclature

| | |
|---|---|
| $\mathcal{G}$ | distribution network graph |
| $\mathcal{N}$ | set of distribution network nodes |
| $\mathcal{E}$ | set of distribution network edges |
| $P_l$ | probability of line failure |
| $\overline{P_l}$ | failure probability under standard weather conditions |
| $P_{l\_hw}$ | line failure probability when faced with high winds |
| $w_{\mathrm{crl}}$ | critical wind speed |
| $w_{\mathrm{cpse}}$ | collapse speed |
| $\mu_j$ | centroid of $j$th cluster |
| $S_L$ | Silhouette index |
| $CH$ | Calinski–Harabasz index |
| $DB$ | Davies–Bouldin index |
| $B_K$ | inter-cluster covariance |
| $W_K$ | intra-cluster covariance |
| $LC_i(j)$ | critical load curtailment of the $j$th reduced scenario for locational combination $i$ |
| $ELC_i$ | expected load curtailment (ELC) for the $i$th locational combination |
| $Pr(j)$ | probability of the $j$th reduced scenario |
| $\omega_x$ | critical load factor at node $x$ |
| $P_{MER-tot}$ | total MER capacity |
| $ELC_{min}$ | minimum ELC for a particular total MER size $P_{MER-tot}$ |
| $t_{outage}$ | duration of power outage |
| $VoLL$ | value of lost load |
| $C_{investment}$ | investment cost of MERs |
| $LCOE_{MER}$ | levelized cost of electricity (LCOE) of MER |
| $t_{backup}$ | backup duration requirement |

## Abbreviations

The following abbreviations are used in this manuscript:

| | |
|---|---|
| MER | Movable Energy Resource |
| ELC | Expected Load Curtailment |
| LCOE | Levelized Cost of Electricity |
| KSFSA | Kruskal's Spanning Forest Search Algorithm |
| NREL | National Renewable Energy Laboratory |
| IEEE | Institue of Electrical and Electronics Engineers |
| PSDR | Power Distribution System Restoration |
| FEMA | Federal Energy Management Agency |

## Appendix A

**Table A1.** Locations of Critical Loads for the 33-node System.

| Nodes | Critical Loads (kW) | Nodes | Critical Loads (kW) |
|---|---|---|---|
| 4 | 60 | 20 | 45 |
| 5 | 30 | 21 | 45 |
| 6 | 60 | 22 | 45 |
| 7 | 200 | 23 | 45 |
| 8 | 200 | 26 | 60 |
| 9 | 60 | 27 | 60 |
| 10 | 30 | 28 | 60 |
| 11 | 25 | 29 | 60 |
| 18 | 45 | 30 | 60 |
| 19 | 45 | 33 | 30 |

**Table A2.** Locations of Critical Loads for the 123-node System.

| Nodes | Critical Loads (kW) | Nodes | Critical Loads (kW) |
|---|---|---|---|
| 1 | 40 | 66 | 75 |
| 6 | 40 | 75 | 40 |
| 11 | 40 | 79 | 40 |
| 17 | 20 | 85 | 40 |
| 24 | 40 | 87 | 40 |
| 30 | 40 | 94 | 40 |
| 37 | 40 | 98 | 40 |
| 43 | 40 | 100 | 40 |
| 50 | 40 | 109 | 40 |
| 52 | 40 | 113 | 40 |

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
