# Peer review of "Optimal Sizing of Movable Energy Resources for Enhanced Resilience in Distribution Systems: A Techno-Economic Analysis"

_electronics, doi:10.3390/electronics12204256_

Round 1

Reviewer 1 Report

The authors address a timely topic that is currently very relevant to industry. Below are some comments for consideration.

- MER sizing is dictated by practical constraints, rather than AI-based optimization. These constraints include size and weight of trailers allowed by DOT, standard offering package sizes, etc. Furthermore, this should not be optimized for a single system (in this case, the 33 bus IEEE test system). Rather than trying to finely pinpoint the exact fine-tuned size, the reviewer recommends working with real life blocks.

- It is not clear to the reviewer why adopting a combination of so many AI-based methods is done. The reviewer recommends comparing the results by both using and not using this fine tune so that the practical aspects of the work can still be used by industry.

- It isn't clear how the road network was used in the optimization. This is a critical element that appears to be missing.

Please proofread it.

Reviewer 2 Report

The aggregate load in IEEE 123 should be indicated, and the load distribution and critical load distribution between the IEEE 30 bus system and the 123 bus system should be displayed and explained in terms of their effects on the final conclusions of the proposed techno-economic analysis on MER should be illustrated and discussed

The quality of writing is good. 

Reviewer 3 Report

The author tackle a modern topic, which is the resilience of distribution grids, and provide interesting results. However, certain comments need to be addressed:

1.      In the Introduction, the authors discuss system restoration strategies using six references. However, since this is the main subject of the manuscript, the authors are encouraged to expand their analysis with recent related research, e.g. such as indicatively:

·         https://doi.org/10.3390/s22239457

·         https://doi.org/10.3390/electronics12153246

2.      The proposed innovation should be highlighted at the end of the Introduction, especially compared to existing solutions/research.

3.      The proposed methodology raises concerns regarding system stability. Presumably, this issue could be overcome with the use of advanced power electronics. Maybe the authors should write a respective comment in the revised version of the manuscript. Relevant literature can be found in:

·         https://doi.org/10.3390/electronics12040931

4.      Regarding Figure 8, could the authors perform a sensitivity analysis? More specifically, how sensitive is the optimal solution, i.e., 1,300 kW, for slightly different parameters? The same question applies to the optimal solution of the 123-node system, presented in Figure 9. This is probably the most important comment and is fundamental for the wider acceptance of the proposed methodology.

5.      How is the proposed methodology validated?

6.      The authors should add a discussion section at the end of the manuscript.

7.      In the Conclusions, it would be useful if not only the power but also the costs of the optimal solutions were mentioned.

8.      The authors could consider adding a nomenclature where all symbols and acronyms are included.

Taking the aforementioned comments into account, a major revision of the manuscript is proposed.

Round 2

Reviewer 3 Report

The authors have addressed the comments adequately. Therefore the manuscript is accepted.